# A Riboswitch-Driven Era of New Antibacterials

**DOI:** 10.3390/antibiotics11091243

**Published:** 2022-09-13

**Authors:** Nikoleta Giarimoglou, Adamantia Kouvela, Alexandros Maniatis, Athanasios Papakyriakou, Jinwei Zhang, Vassiliki Stamatopoulou, Constantinos Stathopoulos

**Affiliations:** 1Department of Biochemistry, School of Medicine, University of Patras, 26504 Patras, Greece; 2Institute of Biosciences & Applications, National Centre for Scientific Research “Demokritos”, Ag. Paraskevi, 15341 Athens, Greece; 3Laboratory of Molecular Biology, National Institute of Diabetes and Digestive and Kidney Diseases, Bethesda, MD 20892, USA

**Keywords:** riboswitch, antibiotics, antibacterial drug targets, RNA

## Abstract

Riboswitches are structured non-coding RNAs found in the 5′ UTR of important genes for bacterial metabolism, virulence and survival. Upon the binding of specific ligands that can vary from simple ions to complex molecules such as nucleotides and tRNAs, riboswitches change their local and global mRNA conformations to affect downstream transcription or translation. Due to their dynamic nature and central regulatory role in bacterial metabolism, riboswitches have been exploited as novel RNA-based targets for the development of new generation antibacterials that can overcome drug-resistance problems. During recent years, several important riboswitch structures from many bacterial representatives, including several prominent human pathogens, have shown that riboswitches are ideal RNA targets for new compounds that can interfere with their structure and function, exhibiting much reduced resistance over time. Most interestingly, mainstream antibiotics that target the ribosome have been shown to effectively modulate the regulatory behavior and capacity of several riboswitches, both in vivo and in vitro, emphasizing the need for more in-depth studies and biological evaluation of new antibiotics. Herein, we summarize the currently known compounds that target several main riboswitches and discuss the role of mainstream antibiotics as modulators of T-box riboswitches, in the dawn of an era of novel inhibitors that target important bacterial regulatory RNAs.

## 1. Introduction

Antibiotic resistance and multiple drug resistance (MDR) are recognized as serious challenges in the management of human health in both community and hospital settings worldwide. There is growing concern that the current antibacterial medication repertoire consists mostly of a narrow set of chemical scaffolds, which target a relatively limited range of biological molecules and functions. This allows microorganisms to bypass antibiotic action mechanisms, leading to MDR, demanding the development of novel antimicrobial agents and therapies [1]. Typically, attempts to discover new treatment possibilities have focused on bacterial proteins as therapeutic targets, with regulatory RNA elements only recently being targeted. The fact that the most commonly used antibiotics target the bacterial ribosomal RNA motivates the development of novel chemical scaffolds that target different biological processes operated by RNA [2].

Riboswitches represent a widespread class of bacterial regulatory RNAs with well-defined structures and critical function in the regulation of key genes [3]. They are commonly found in bacteria and are almost exclusively located in the 5′ UTRs of mRNAs to control the expression of downstream coding regions [4,5]. They fold into complex tertiary structures and consist of a ligand-binding domain (aptamer) that selectively binds cognate small molecule ligands such as nucleotides, amino acids, vitamins, etc. This, in turn, induces a conformational change to the regulatory domain (expression platform), affecting either the transcription or the translation of the regulated genes [6,7]. Commonly, this ligand-induced conformational change leads to the formation of a terminator hairpin that halts RNA polymerase, or of a hairpin that sequesters the Shine–Dalgarno sequence and prevents the ribosome from binding and initiating on the mRNA, or both, leading to the switching-off of the downstream gene. This group of riboswitches are described as “off”-switches. On the other hand, in the “on”-switches, the ligand-induced conformational switch results in the formation of an anti-terminator stem or the release of the RBS from sequestration, which promotes the transcription or translation of the downstream gene, respectively, and activates latent gene expression from the adjacent coding region [8,9] (Figure 1). They affect bacterial homeostasis, development, pathogenicity and antibiotic resistance at the transcriptional or translational level and can create complex three-dimensional structures with well-defined small-molecule binding sites, posing them as attractive antimicrobial therapeutic targets [10,11,12].

Approximately 40 different classes of riboswitches have been identified and validated so far, with some classes being ubiquitous, present in nearly all lineages of bacteria, such as the TPP riboswitch, while others can be extremely rare, such as the preQ1-III riboswitch [13]. They are classified according to their cognate ligand aptamer domain which possesses highly selective binding sites for ligands and is highly conserved among different species [6]. Their natural ligands include coenzymes (flavin mononucleotide (FMN), thiamin pyrophosphate (TPP), coenzyme B12); nucleobases and their derivatives (adenine, guanine, c-di-GMP, c-di-AMP, pre-queuosine (preQ1)); cations such as Mg^2+^ and Mn^2+^, Ni^2+^/Co^2+^ and Na^+^ and anions such as fluoride, amino acids and derivatives (lysine, glycine, S-adenosyl methionine (SAM), S-adenosyl homocysteine (SAH)); carbohydrates (glucosamine-6-phosphate (GlcN6P)); and guanidine [14,15,16,17,18,19,20,21,22]. T-box riboswitches represent a distinct class of riboswitches whose cognate ligand is the tRNA molecule, which interact through a codon–anticodon-like manner, and are involved in amino-acid metabolism [23,24,25].

Several lines of genetic evidence support riboswitches as key players in the regulation of bacterial vital pathways through feedback control mechanisms and as a result, as potential and novel high-potential targets for antibiotic design to combat MDR. For instance, some riboswitches that are widely distributed among different bacterial species (i.e., TPP, FMN riboswitches) could be used as broad-spectrum antibacterial targets, while others that are found specifically in individual genus or species could constitute selective drugs (e.g., targeting preQ-1 or purine-riboswitches) [26]. Recently, it was reported that T-boxes in *Staphylococcus aureus* contain species-specific structural features that play a central role in riboswitch structural conformation and function, allowing the adaptation to specific metabolic environments, making them promising, lineage-specific drug targets [27,28,29,30]. Furthermore, some riboswitches regulate the expression of several genes that are fundamental for bacterial survival. In *Salmonella typhimurium*, a coenzyme B12 (adenosylcobalamin, AdoCbl) riboswitch regulates the expression of the *btuB* gene, which encodes a cobalamin transport protein, whereas a second riboswitch of the same class acts as a genetic control element for the cob operon, which contains 25 genes whose collective suppression could prevent AdoCbl biosynthesis and transport. Targeting this riboswitch could lead to the deprivation of AdoCbl, which can halt bacterial growth [31]. Moreover, an aminoglycoside-sensing riboswitch was proposed to control the expression of the aminoglycoside acetyl transferase (AAC) and aminoglycoside adenyl transferase (AAD) genes, both of which confer resistance to aminoglycosides in bacteria. Notably, this riboswitch was suggested to sense and bind aminoglycosides, inducing the expression of the aminoglycoside resistance genes, leading to antibiotic resistance [32,33]. Therefore, riboswitches that sense and bind antibiotics might constitute a powerful strategy against antibiotic resistance in pathogenic bacteria.

The design of efficacious riboswitch-targeting compounds must comply with specific requirements depending on the targeted riboswitch class, the compound uptake mechanism and the pathway stimulated or inhibited that will lead to the antimicrobial effect. Thus, the targeted riboswitch should control a gene that is essential for the bacterial growth or pathogenicity. Another crucial criterion for developing an effective antimicrobial compound is that the lead compound does not functionally substitute for the natural ligand for its intracellular activity, which could result in unwanted intracellular off-targets effects in the infected host cells [26]. Overall, a thorough understanding of the riboswitch architectures and the pre-conditions required for their conformational switch is of high significance in order to achieve a targeted design of efficient compounds that bind specifically to riboswitches and induce the appropriate folding changes that result in cell growth arrest or killing.

In the present review, we highlight a select group of newly identified and validated, promising compounds which target riboswitches and appear as suitable lead compounds for the development of a new generation of antibacterial drugs. We also discuss the role of mainstream antibiotics on the function of known riboswitches, such as the T-boxes, in prominent human pathogens. Given the urgent need for new, specific and robust therapies to combat bacterial infections, the development of new inhibitors that target essential bacterial RNA regulatory elements holds great promise for the future.

## 2. TPP Riboswitches

The thiamine pyrophosphate (TPP) riboswitch is the most widespread riboswitch present in all three domains of life, archaea, bacteria, and eukaryotes, including algae [34], fungi [35] and plants [36]. It resides in the 5′ UTR of important genes involved in biosynthesis and the transportation of TPP, such as *thiMD*, *thiCEFSGH* and *thiBPQ* genes in *Escherichia coli*, *nmt1* in *Neurospora crassa* and *AtTHIC in Arabidopsis thaliana*. Strikingly, a TPP riboswitch was recently identified in Arabidopsis, which in contrast to its prokaryotic and fungal counterparts is located in the 3′ UTR of the plant thiamine c synthase gene (*AtTHIC*) [36,37,38,39,40]. TPP, also named thiamine diphosphate (ThDP) or cocarboxylase, is the biologically active form of thiamine (Vitamin B1) and it is the natural ligand of TPP riboswitches. Structurally, TPP consists of an aminopyrimidine ring linked to a thiazolium ring with a pyrophosphate tail [41]. Another natural compound that can bind to TPP riboswitches via its aminopyrimidine ring is thiamine monophosphate (TMP). However, TMP exhibits lower binding affinity to the riboswitch compared to TPP due to the lack of pyrophosphate contacts [42].

The crystal structure of the *thiM* riboswitch from *E. coli* bound to TPP revealed a Y-shaped riboswitch consisting of two parallel helical domains connected by a P1 stem helix, and a TPP molecule bound between them. The helical domains consist of P2/J3-2/P3/L3 and J2-4/P4/P5/L5 small helices, respectively. The aminopyrimidine ring of TPP binds to the P2/P3 region, while the central thiazolium ring of TPP does not appear to be bound to any region of the riboswitch. In addition, the pyrophosphate tail is bound with two magnesium ions (Mg^2+^), which in turn interact with the P4/P5 helices [42,43]. These Mg^2+^ are required for the anchoring of TPP to the riboswitch [44]. The crystal structures of TPP riboswitches from other organisms of different domains of life, such as the TPP riboswitch of *thiC* gene from *A. thaliana*, revealed very similar folding [45], suggesting that due to their highly conserved TPP recognition mechanism, the synthetic, new ligands that mimic TPP will likely bind multiple TPP riboswitches from different organisms.

TPP riboswitches can regulate gene expression in *cis*, both at transcription and at translation level, through their conformational change between an “on” state in low TPP concentration and an “off” state in high TPP concentration [43]. Interestingly, it has been shown in eukaryotes that the TPP riboswitch can cause the alternative splicing of mRNA in the presence of its ligand [46]. Overall, the widespread presence of the TPP riboswitch in a large variety of human pathogens such as *S. aureus*, *H. pylori*, *N. gonorrhoeae*, *S. pneumoniae*, *H. influenzae* and *M. tuberculosis*, together with its multilevel regulation of gene expression make it a promising target for new broad-spectrum antibacterial compounds [47].

One of the first thiamine analogs that was studied was oxythiamine, in which a hydroxyl group replaces the exocyclic amino group of its pyrimidine ring [38]. In the cell, this analog is phosphorylated and turns into oxythiamine pyrophosphate (OTPP). It has been shown that oxythiamine displays a strongly reduced binding affinity to the 165 nucleotide *thiM* RNA fragment of *E. coli* because it fails to induce structure modulation compared with the TPP ligand [38]. However, its antifungal activity was revealed later, once oxythiamine was shown to decrease both the growth rate and survival of *Saccharomyces cerevisiae* [48]. Another two synthetic thiamine derivatives with distinct structural characteristics from TPP, benfotiamine (BFT) and amprolium, were also studied for their binding affinity to the 165 *thiM* riboswitch fragment of *E. coli*, but failed to induce any structural rearrangements in the riboswitch [38].

A compound that established the TPP riboswitch as an antibacterial drug target is pyrithiamine (PT), a synthetic analog of thiamine in which the thiazole ring is replaced by a pyridinium ring [49]. Interestingly, this compound was designed for thiamine metabolic studies, well before the TPP riboswitch was discovered (Table 1). PT can be taken up by cells via thiamin transporters and subsequently phosphorylated by thiamine pyrophosphokinase to form pyrithiamine pyrophosphate (PTPP), the active form of PT. In vitro studies in *Bacillus subtilis* and *Aspergillus oryzae* have revealed that the antimicrobial mechanism of PT is based on its binding to TPP riboswitches of these organisms with nearly identical affinities as TPP, leading to the repression of riboswitch-regulated thiamin biosynthesis and transport genes. These results demonstrate that this riboswitch can act as a potential target for structure-based design for both antibacterial and antifungal agents. However, PT-resistant bacteria have already been emerged and mutations within the TPP riboswitch have been identified. These mutants block ligand binding to the aptamer domain, and as a result there is a deregulation of the genes involved in TPP metabolism [49].

A series of synthetic TPP analogs with modifications at either the aminopyrimidine ring, the thiazole ring, or the pyrophosphate moiety of TPP have been tested as possible inhibitors of the *E. coli thiM* riboswitch [50]. Biophysical binding experiments using [^3^H] thiamine-dependent equilibrium dialysis indicated that the most efficient compounds were 2,4-diaminopyrimidine (8T), methylenediphosphonate [20] and N-phosphosulfamate [21]. Further studies with these compounds investigated whether any of these analogs could repress the gene expression of the luciferase gene under the expression control of the TPP riboswitch by using an in vitro transcription/translation (IVTT) assay. Only compound 20 was able to repress expression to the same extent as TPP (relative luminescence of 0.40 versus 0.39 for TPP). Thus, it was concluded that the pyrophosphate moiety is essential for submicromolar binding affinity, but unexpectedly, it does not appear to be strictly necessary for the modulation of gene expression [50].

An additional study reported the design of several compounds containing a central 1,2,3-triazole group instead of the thiazole group presented in TPP [51]. The compound that induced the inhibition of *thiM*-riboswitch in different *E. coli* strains was triazolethiamine (TT), which requires phosphorylation by thiamin kinase (ThiK) to be activated. The active form of TT is called triazolethiamine pyrophosphate (TTPP), and its binding affinity to the TPP riboswitch is 370 nM compared to 8 nM for TPP. The same study reported that TTPP binding to the TPP riboswitch causes changes to the secondary structure of the riboswitch and as a result, it sequesters the SD-sequence from binding to ribosome, leading to inefficient protein translation. To bypass the ThiK dependency of TT, derivatives of TT containing phosphate mimics such as sulfate, sulfonamide, or sulfone were designed. Among them, only one compound [7] showed a significant inhibition of thiM-riboswitch expression [51].

Recently, two more compounds, CH2-TPP and CF2-TPP, have been designed as *E. coli thiM* riboswitch ligands, and their binding affinity was estimated using well-tempered metadynamics (WT-MtD) simulations. The results revealed that between the two ligands, CH2-TPP showed higher binding affinity and repressed gene expression to the same extent as upon high concentrations of TPP [52]. However, these compounds have not been studied further for their potential antimicrobial activity.

**Table 1 antibiotics-11-01243-t001:** List of the compounds that target a specific class of riboswitches and exhibit antimicrobial activity for each of the indicated organisms.

Riboswitch	Antimicrobial Compound		Organism	Ref.
TPP	Pyrithiamine	* 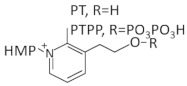 *	*B. subtilis,* *A. oryzae*	[49]
FMN	Roseoflavin	* 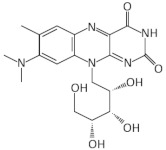 *	*L. monocytogenes,* *B. subtilis,* *E. faecalis,* *S. pyogenes*	[53,54,55]
	5FDQD	* 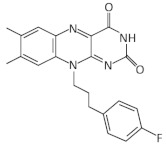 *	*B. subtilis,* *C. difficile*	[56,57]
	Ribocil	* 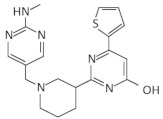 *	*E. coli*	[58]
	Ribocil-C	* 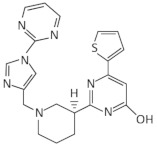 *	*E. coli,* *S. aureus*	[58,59]
	Ribocil C-PA	* 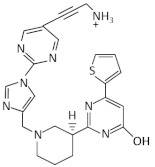 *	*E. coli,* *K. pneumoniae*	[60]
	10-(2,2-dihydroxylethyl)-7,8-dimethylisoalloxazine (5a)	* 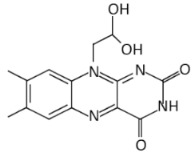 *	*M. tuberculosis*	[61]
	ASO-1	T_1_T_1_C_1_T_2_C_2_C_2_C_2_A_2_T_2_C_2_C_2_A_2_G_2_A_1_C_1_T_1_	*S. aureus,* *L. monocytogenes,* *E. coli*	[62]
GlmS	ASO1	C_1_T_1_T_1_T_2_A_2_A_2_C_2_T_2_G_2_T_2_A2C_2_T_2_G_1_C_1_C_1_	*S. aureus*	[63]
	carba-α-D-glucosamine	* 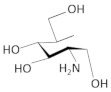 *	*S. aureus*	[64]
	carba-α-D-glucosamine-6-phosphate	* 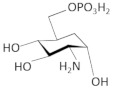 *	*S. aureus*	[64]
	fluoro-carba-α-D-glucosamine-6-phosphate	* 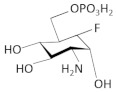 *	*B. subtilis,* *S. aureus*	[65]
Guanine	PC1	* 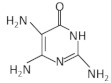 *	*S. aureus,* *C. difficile,* MDR strains	[66,67,68]
T-box	Neomycin B	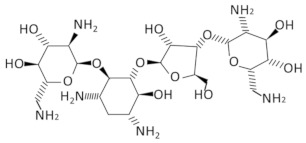	*B. subtilis,* *S. aureus*	[28,69]
	PKZ18	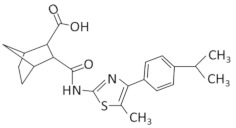	*B. subtilis,* *S. aureus*	[70]
	PKZ18-22	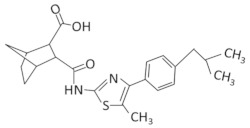	*B. subtilis,* *S. aureus,* MRSA	[71,72]

1 = 2′-alkyl modifications of the ribose; 2 = phosphorothioate (PS) linkage.

## 3. FMN Riboswitches

FMN riboswitches are widespread among bacterial species and regulate the biosynthesis and transport of the vitamin riboflavin. Riboflavin is the precursor of flavin mononucleotide (FMN), the availability of which regulates the conformational switch of the riboswitch and in particular, of its aptamer domain. Structurally, a FMN riboswitch consists of six helices that form a butterfly-like structure. Its FMN ligand is directed in the center of the junctional region and one of its four oxygens is used for hydrogen bonding, whereas phosphate oxygens interact with several conserved guanines, such as G33 [73]. On the other hand, riboflavin which lacks the phosphate exhibits 1000-fold lower binding affinity to the FMN aptamer of *F. nucleatum* and *B. subtilis* [74,75,76].

The FMN riboswitch is located in the 5′ untranslated region of the *ribD* operon, which consists of *ribD*, *ribE*, *ribA*, *ribH*, and *ypzK/ribT* genes required for riboflavin biosynthesis. In the absence of FMN, the operon is fully transcribed, whereas in the presence of FMN it binds to the riboswitch and leads to the formation of the terminator stem, resulting in the premature transcription termination of the *ribD* operon mRNA [74,77,78]. Gram-positive bacteria are also able to import exogenous riboflavin in contrast to Gram-negative bacteria. The expression of the relative riboflavin transporter, *ypaA*, is regulated by a FMN riboswitch at the translational level. In this case, the presence of FMN prevents the interaction of the small ribosomal subunit with the ribosome binding site (RBS). By contrast, FMN deficiency enables the translation of the *ypaA*, as the RBS is more flexible and accessible to the ribosome [79]. As riboflavin is a metabolite involved in the key cellular pathways required for bacterial physiology, targeting the FMN riboswitch with FMN analogs can lead to bacterial growth inhibition [80].

Roseoflavin is a natural analog of riboflavin produced by *Streptomyces davawensis* [81]. Upon the phosphorylation of roseoflavin (RoFMN) it binds to the FMN riboswitch with similar affinity to FMN and prevents the expression of downstream genes [73] (Figure 2). RoFMN exhibits antimicrobial activity against both Gram-negative and Gram-positive bacteria, including *L. monocytogenes*, *B. subtilis*, *E. faecalis* and *S. pyogenes* [53,54,55] (Table 1). However, the requirement of roseoflavin activation through phosphorylation by bacterial enzymes, such as RibC in *B. subtilis*, limits its antimicrobial activity [82,83]. In addition, roseoflavin has low selectivity, as it targets many different flavoproteins that are conserved in humans, thus, limiting its potential as an FMN riboswitch-specific antibiotic [84,85,86].

With the aim of discovering RoFMN analogs that exhibit antimicrobial activity but do not require activation for their transport and phosphorylation for their activity, a structure-based approach was employed [56,57,87,88]. This study led to the discovery of a series of compounds that bind to the FMN riboswitch in vitro, which were patented (P. D. G. Coish et al., 20 January 2011, patent application PCT/US2010/001876; P. D. G. Coish et al., 13 October 2011, patent application PCT/US2011/000617; P. A. Aristoff, P. D. G. Coish, and B. R. Dixon, 12 August 2012, patent application PCT/US2012/024507). The first analog, BRX830, was produced by removing the hydroxyl groups at position N10 of RoFMN, which do not interact with the aptamer. It was shown that this analog could still bind efficiently to the riboswitch in vitro. BRX830 was further modified by replacing the phosphate group with a carboxylic group and by adding a cyclopentyl ring at position N8, leading to BRX1151 that retained both the high binding affinity of BRX830 to the FMN riboswitch and exhibited very low IC_50_ and EC_50_ compared to the other candidates. Further modifications of these analogs resulted in 5FDQD (5-(3-(4-fluorophenyl)butyl)-7,8-dimethylpyrido [3,4-b]quinoxaline-1,3(2H,5H)-dione), which differ substantially from FMN since it bears a N1-deaza flavin core and an aryl-alkyl moiety instead of a ribityl phosphate moiety of FMN [56,57]. In vitro studies using the *ribD* FMN aptamer of *B. subtilis* have shown that 5FDQD and FMN have very similar binding affinities, with K_D_ values of 7.5 nM and 6.4 nM, respectively. In addition, 5FDQD targets the FMN riboswitch of *Clostridium difficile* and exhibits antimicrobial activity with a MIC_90_ of 1 μg/mL, similar to vancomycin, metronidazole and surotomycin, the latter of which is under phase III clinical trial investigation for *C. difficile* infection (CDI) therapy [56,57,87,88]. 5FDQD is also effective for the treatment of the antibiotic-induced CDI in mice, similar to fidaxomicin, which is an established antibiotic for CDI with limited side effects in normal cecal flora compared to fidaxomicin and vancomycin. Overall, the neutral form of 5FDQD that facilitates cell uptake, its high binding affinity for the FMN aptamer without the requirement of activation, as in case of RoFMN, and its low propensity to develop resistance (<1 × 10^−9^ frequency) makes it a promising agent. However, 5FDQD’s exact mechanism of action and target specificity for the FMN riboswitch is still under investigation [56,57].

A synthetic compound with antimicrobial activity, ribocil, which is able to target the FMN riboswitch, has been identified using a phenotypic screen [58]. Ribocil inhibited the bacterial growth of *E. coli* MB5746, which is defective in wild-type lipopolysaccharide (LPS) levels and drug efflux. A dose-dependent decrease in riboflavin, FMN and FAD levels was detected in the presence of ribocil, whereas its inhibitory effect was totally suppressed in the presence of exogenous riboflavin. Notably, all the ribocil-resistant mutants were located on the FMN riboswitch, indicating the selective inhibition of the FMN riboswitch-regulated riboflavin biosynthesis by ribocil. Its co-crystallization with the *Fusobacterium nucleatum impX* FMN riboswitch revealed that ribocil and FMN share conserved contacts with the aptamer, including G11 and G62, which interact with the phosphate group of FMN. Although ribocil is a mixture of two isomers (ribocil-A and ribocil-B), its structural characterization revealed that only the S enantiomer, ribocil B, binds to the aptamer. In addition, ribocil-B exhibited lower MIC and IC_50_ values, as well as higher binding affinity to the *E. coli* FMN riboswitch (K_D_ = 6.6 nM) with respect to ribocil-A (K_D_ ≥ 10,000 nM), confirming that only ribocil-B binds to the riboswitch. Ribocil-C, an analog of ribocil-B with eight-fold higher antibacterial activity than ribocil, was tested against murine *E. coli* septicemia and exhibited great efficiency in clearing murine infection [58,59]. Ribocil-C displayed antibacterial activity towards methicillin-resistant *S. aureus* (MRSA) (MIC = 0.5 mg/mL), which was reduced in the presence of riboflavin in vitro. Ribocil-C selectively binds both *S. aureus* FMN riboswitches for riboflavin biosynthesis and riboflavin uptake with a similar binding affinity with FMN. Notably, in the presence of ribocil, FMN riboswitch located upstream of the *ypaA* gene was detected to undergo ribocil resistance mutations [55]. Being more potent for Gram-negative bacteria, ribocil was further modified to carry a primary amine (ribocil C-PA), which favored its accumulation inside the bacteria and showed high activity against *E. coli* and *K. pneumoniae* in vivo [60].

In a recent study, many riboflavin derivatives were tested for their antimicrobial activity against *M. tuberculosis*. The most potent was compound 5a (10-(2,2-dihydroxylethyl)-7,8-dimethylisoalloxazine), which displayed antimicrobial activity against *M. tuberculosis* H37Rv with a MIC_99_ of 6.25 μM. Binding experiments in conjunction with docking calculations suggested that 5a can bind to the FMN aptamer and inhibits riboflavin’s biosynthesis, leading to bacterial growth arrest [61].

A different approach to identifying new FMN-riboswitch ligands was carried out using the Automated Ligand Identification System (ALIS), a label-free AS-MS, in which ~53,000 compounds were screened similarly to the earlier identification of ribocil. Τwo compounds that interacted with an FMN riboswitch with high affinity were identified (WG-1 and WG-3), albeit their inhibitory activity on the bacterial growth was independent of the riboflavin pathway [89]. Another technology that facilitates the screening of billions of compounds simultaneously in a single reaction is the DNA-encoded library (DEL); however, it is not well established for RNA targets yet, although one successful attempt has identified two compounds, HGC-1 and HGC-2, with binding affinities for *E. coli* FMN riboswitch of 11.38 and 15.03 nM, respectively. Nonetheless, these compounds must be further investigated for their ability to penetrate the bacterial cell wall and antimicrobial activity [90].

The use of antisense oligonucleotides (ASOs) that target a specific riboswitch as antimicrobial agents was first applied to target the *glmS* riboswitch/ribozyme [63]. The same strategy led to the discovery of the antisense oligonucleotide ASO-1, which hybridizes with its complementary FMN riboswitch aptamer. Specifically, ASO-1 contains a sulfur atom that has replaced an unbound oxygen and targets a conserved region of the FMN riboswitches that regulate either the *ribD* operon or *ypaA* gene. ASO-1 is delivered into the bacterial cells by a non-toxic oligopeptide (pVEC) derived from the murine vascular endothelial-cadherin protein, which is degraded by peptidases after its entry into the cells. The ASO-1/FMN aptamer hybrid is recognized and cleaved by RNase H. Strikingly, ASO-1 at the same concentration showed a bacteriostatic effect against *S. aureus*, *L. monocytogenes*, and *E. coli* with MIC_80_ 700 nM and non-toxic effect in the lung cancer A549 cell line [62]. Antisense oligonucleotides seem to be promising antimicrobial agents due to their straightforward design, as well as their convenient cellular delivery. Another major advantage is that they can be designed to target regions with a low mutation rate, which is indispensable for riboswitch function. New oligonucleotide analogs will be able to target the riboswitches of many other resistant bacterial strains and overcome the antimicrobial resistance, which has become a global problem due to the widespread misuse and overuse of antibiotics.

## 4. GlmS Riboswitch-Ribozyme

The *glmS* riboswitch/ribozyme is found in Gram-positive bacteria and belongs to an unusual class of riboswitches due to its catalytic RNA activity, acting as a nucleolytic self-cleaving ribozyme. Its cognate ligand is glucosamine-6-phosphate (GlcN6P), an essential precursor for the bacterial cell wall formation [91,92]. The *glmS* riboswitch resides in the 5′ UTR of the glutamine-fructose-6-phosphate aminotransferase (*glmS*) gene, which plays a key role in GlcN6P biosynthesis. It catalyzes the first step in hexosamine metabolism, converting fructose-6P into GlcN6P by using glutamine as a nitrogen source [93]. Under a sufficient concentration of GlcN6P, the natural ligand induces the riboswitch to self-cleave the downstream ORF of *glmS* mRNA in two specific fragments, thereby reducing *glmS* expression. The first, 5′ fragment has a 2′,3′-cyclic phosphate end (2′,3′-cP), while the second, 3′ fragment has a 5′-OH end [94]. It has been suggested that the latter fragment is rapidly degraded in *B. subtilis* by RNase J1, which means that GlcN6P is responsible for the *glmS* mRNA degradation [95]. Evolutionary studies have revealed that the catalytic part of the *glmS* riboswitch is >97% identical across bacterial species, which makes the *glmS* riboswitch a promising target for antimicrobial drugs for human pathogens [96].

In order to examine the essential requirements for ligands to induce *glmS* ribozyme self-cleavage, a great number of amine-containing analogs was used, revealing that the amine functional group of GlcN6P is essential for ribozyme catalysis [97]. This observation was later confirmed when the structure of the *glmS* ribozyme was first determined using X-ray crystallography [98]. By examining the structure of the riboswitch in a pre-cleavage state, metabolite-bound state and post cleavage state, no conformational change was observed. Thus, it was suggested that the ligand binds to a preorganized active site in the ribozyme that depends on GlcN6P as a cofactor, not as an allosteric ligand activator. The crystal structure of the *glmS* riboswitch also revealed that it consists of three coaxial stacks of RNA helices packed side by side. The longest stack (~100 Å long) is surrounded by a P1 loop and a P3.1 helix. This stacks together with a smaller stack enclosing a short central stack, P2.1. The riboswitch’s core, which forms a double pseudoknot structure, is positioned in the central stack. This structure suggests that the amine group of GlcN6P is essential to perform general acid-base and electrostatic catalysis [98].

After the structural analysis of *glmS* riboswitch, many structure–activity relationship (SAR) studies have been conducted to explore new compounds that could target this riboswitch [26,99]. A high-throughput assay (HTS) based on fluorescence resonance energy transfer (FRET) for screening GlcN6P analogs that bind to and activate *glmS* ribozyme cleavage was developed. Using a library of 960 compounds previously approved for use in humans, five active compounds were identified. Among them, glucosamine stimulated detectable cleavage, while glucose-6-phosphate (Glc6P) and glucose (Glc) could not, suggesting the importance of the phosphate moiety of GlcN6P and confirming again the essential role of the amine group [26,99]. These results reveal the essential moieties a future ligand must have in order to target the *glmS* riboswitch.

Other interesting findings have emerged using a fluorescence polarization-based assay, where up to 5000 Glc6NP analogs and numerous 1-deoxy, 1-methyl glycoside, and carba-sugar derivatives of Glc6NP were screened [100]. None of the 1-deoxy and 1-methyl glycoside derivatives, nor the 5000 from the former library were active. However, the carba-sugar derivatives activated the *glmS* riboswitch to self-cleave. Thus, it was concluded that the ring oxygen is dispensable for the ribozyme activity [101]. In 2017, the same group revealed the antibacterial activity of carba-α-D-glucosamine (CGlcN) and carba-α-D-glucosamine-6-phosphate (CGlcN6P) in *S. aureus* [64] (Table 1). They also examined mono-fluorinated carba-derivatives of GlcN6P and concluded that fluoro-carba-α-D-glucosamine-6-phosphate was the most effective compound, which activated the *glmS* ribozyme in both *B. subtilis* and *S. aureus*. However, all the fluorinated carba-derivatives were less active compared to the non-fluorinated analogs of the previous study [65]. Lastly, Soukup’s group designed many phosphate analogs of GlcN6P. Among these, only two exhibited a potent activation of the ribozyme activity, namely 6-deoxy-6-phosphonomethyl and 6-O-malony-ether analogs [102].

Although many previously designed GlcN6P-analogs activate the *glmS* ribozyme, only a few of them exhibited an antibacterial effect or target selectivity, which reflects the limited diversity of the currently available chemical libraries. Besides small-molecule compounds, the abovementioned alternative strategy that has recently gained attention is the use of antisense oligonucleotides. A recent study used a chimeric antisense oligonucleotide (ASO1) that was designed against the *glmS* riboswitch of *S. aureus* [63]. ASO1 was modified in the alkyl group of the ribose 2′ position to bear a phosphorothioate group (PS) in the phosphodiester bond. To enter the cell wall, ASO1 5′-end was attached to the pVEC, similar to the ASO-1 that was designed for the FMN riboswitch. The outcome of this study showed that ASO1 specifically targeted the *glmS* riboswitch of *S. aureus* and inhibited its growth with a minimum inhibitory concentration for 80% inhibition (MIC_80_) of 5 μg/mL, while it did not have any effect on the growth of *E. coli* and *B. subtilis* [63]. Taken together, developing antibacterial agents against the *glmS* riboswitch/ribozyme is of great importance due to the essential role of this riboswitch to regulate the synthesis of a crucial precursor of the bacterial cell wall, GlcN6P.

## 5. Guanine Riboswitches

Guanine riboswitches regulate the biosynthesis and transport of purine nucleotides [103]. The three-dimensional structure of the guanine riboswitch is well characterized in many organisms including *B. subtilis*. The guanine riboswitch is organized in a three-way junction stabilizing the P1, P2, P3 helices. P1 functions as an anti-antiterminator and P2-P3 form a loop–loop interaction to enable ligand binding. When in abundance, guanine binds to the aptamer and leads to transcription attenuation. In the absence of guanine, the 3′ strand of P1 interacts with the expression platform and forms the antiterminator structure, enabling the transcription of the downstream genes [104]. Although purine riboswitches are very similar, guanine is strongly favored (10,000-fold) over adenine by forming a Watson–Crick base pair with a single pyrimidine (Y74), and is also distinguished from 2′-deoxyguanosine by U51 that selectively binds a nucleobase [105,106]. However, guanine riboswitch variants have also been detected to sense adenine, xanthine and 2′-deoxyguanosine [107].

The first structure-based approach for the design of compounds targeting guanine riboswitch in *B. subtilis* was accomplished from the group of Breaker. *B. subtilis* has four conserved guanine riboswitches regulating the metabolism of *xpt-pbuX*, *pbuG*, and *yxjA* and the *pur* operon that is required for purine biosynthesis [108]. Their attempts to simultaneously delete all these genes were unsuccessful and gave rise to the idea that a compound specifically targeting the four guanine riboswitches would be lethal for the bacteria. In addition, they observed that the chemical modifications of C2 and C6 of guanine would not affect its interaction with the aptamer, thus, 16 analogs were synthesized and tested for bacterial growth inhibition. Most of these compounds exhibited high binding affinity to the *xpt-pbuX* riboswitch in vitro using in-line probing, even higher than guanine, and inhibited bacterial growth. Analog G7 (6-N-hydroxylaminopurine) was the most effective in inhibiting cell growth with a relatively high MIC (260 μΜ), and repressed the expression of a reporter gene under the control of guanine riboswitch at this concentration in minimal medium [109,110]. The same analog was also reported to target the guanine riboswitches of the Gram-positive *L. monocytogenes* and inhibit its bacterial growth. However, it was revealed that its activity does not depend on the inhibition of the riboswitch-regulated genes, but on the downregulation of the virulence factor PrfA and the increase in the mutation rate and the SOS response [111]. Remarkably, N2-phenoxyacetyl guanine and N2-isobutyrylguanine modified on C2 position of guanine presented a similar binding affinity to guanine for the *B. subtilis* guanine riboswitch, resulting in transcription termination. However, the inhibitory activity of these compounds on the bacterial growth has yet to be determined [112].

The same strategy was also followed to design pyrimidine analogs that bind to the guanine riboswitch and have antimicrobial activity in pathogenic bacteria. Two pyrimidine, non-ribosylable compounds, PC1 (2,5,6-triaminopyrimidin-4-one) and PC2 (2,6-diaminopyrimidin-4-one), were identified to fit properly in the guanine riboswitch. An in-line probing analysis showed that although they both interact with the riboswitch, PC1 displayed more interaction sites than PC2 [66]. PC1 exhibited a lower binding affinity of ~100 nM, similar to hypoxanthine, compared to ~5 nM by guanine. PC1 inhibited *S. aureus* growth in a dose-dependent manner in rich medium and did not result in resistant bacteria even after 30 bacterial passages. Bacterial inhibition was also observed in nine out of 15 Gram-positive bacterial species that were tested, including *C. difficile* and MDR strains (Table 1). Noticeably, a guanine riboswitch that regulates the expression of the guanosine monophosphate (GMP) synthetase, encoded from *guaA*, was detected in all the susceptible species. Further supplementation with GMP resulted in a significant reduction in the bacterial inhibitory activity of PC1, indicating the *guaA* specificity of PC1. Its bactericidal activity was further tested in a murine model of *S. aureus*-induced mastitis where the therapeutic effect of the compound was similar to known antibiotics [66]. PC1 reduced the *S. aureus* levels in *S. aureus*-induced bovine intramammary infection (IMI) in dairy cows and managed to totally cure 15% of them [67]. Nevertheless, the absence of bacterial resistance in combination with the maintenance of a percentage of the inhibitory activity of PC1 in the presence of GMP necessitates the evaluation of its specificity to the guanine riboswitch [66,68].

## 6. Cyclic-di GMP Riboswitches

Cyclic-di-GMP riboswitches consist of two classes, termed class I (c-di-GMP-I) and class II (c-di-GMP-II), which have developed different mechanisms for c-di-GMP recognition and require different structural characteristics of their ligand binding and specificity [15,113,114]. The c-di-GMP responsive riboswitch (GEMM motif) is located upstream of the genes that are involved in c-di-GMP metabolism, as well as the genes that are implicated in pathways regulated by this ligand. In different bacterial species, including several pathogens such as *Clostridium difficile*, *Vibrio cholerae* and *Bacillus anthracis*, over 500 cases of class I c-di-GMP-binding riboswitches and 45 examples of class II riboswitches have been found, with a few examples of both classes present in a single organism [15,113]. By interfering with transcription or translation initiation, the class I c-di-GMP riboswitch modulates gene expression. Riboswitches of this type can be “off”-switches, lowering gene expression at high c-di-GMP levels, or “on”-switches, promoting gene expression upon ligand binding, depending on the gene regulated [15]. Class II cyclic-di-GMP riboswitches were identified to be allosteric ribozymes. When cyclic-di-GMP is bound in the presence of GTP, it causes a rise in spliced exons, creating a functional Shine–Dalgarno sequence that permits translation initiation [113]. As a result, members of this class tend to be “on”-switches. Furthermore, as the spliced RNA is still able to respond to cyclic-di-GMP by exposing or blocking the newly produced SD-sequence, it offers a second level of regulation [115].

After obtaining their crystal structures, each class was reported to have distinct structural features. Regarding class I cyclic-di-GMP riboswitches, three helices form a Y-shape [116,117,118], while class II riboswitches fold in a more rigid structure containing a kink-turn and a pseudoknot [114]. When bound to both riboswitches, cyclic-di-GMP is incorporated either in a double helix in class I or a triple helix in class II, contributing to the extremely tight binding affinities observed (10 pM to low nanomolar), making it the tightest c-di-GMP receptor and the highest known affinity of RNA–ligand interaction [15,113,116]. The ligand is bound to the class I cyclic-di-GMP riboswitch by Gβ, which forms a Watson–Crick base pair with a highly conserved cytosine of the aptamer domain [116], and Gα, which is recognized by a guanine from the aptamer domain. In addition, a highly conserved adenosine intercalates between the ligand’s two bases, resulting in stacking interactions that connect the riboswitch’s P1 and P2 helices [116]. The phosphodiester backbone is also recognized by the class I riboswitch. Metal coordination, hydrogen-bonding interactions to the phosphates and ribose 2′ hydroxyls all contribute to c-di-GMP binding. The recognition of the ligand by the class II riboswitches depends on base-stacking interactions [114]. The guanine bases are again asymmetrically recognized but without any canonical pairing. Gβ is recognized as part of a base triple and forms two hydrogen bonds with the riboswitch’s aptamer domain and a hydrated magnesium ion [114].

A group of four different circular cyclic-di-GMP analogs (2: cyclic bis(3′-5′)-2′-deoxyguanylic/guanylic acid (cdGpGp); 3: monophosphothioic acid of c-di-GMP (c-GpGps); 4: cyclic bis(3′-5′)guanylic/adenylic acid (c-GpAp); and 5: 2′-O-di(tert-butyldimethylsilyl)-c-di-GMP (2-OTBDMS CDG)) were tested for their affinity to bind to the class I and class II aptamers and showed that the class I aptamer retains a substantial affinity (*K*_D_ ≤ 10 nM) with compounds 2 and 3 that bear minor modifications. On the other hand, compound 4, in which a single GMP is substituted by AMP, and compound 5, in which the 2′-OH group is substituted by a bulky tert-butyl dimethylsilyl group (TBDMS), exhibited significantly reduced affinities to the aptamer. The class II aptamer, in contrast to the class I, retains its high affinity with all tested circular analogs (*K*_D_ ≤ 10 nM). According to structural models, class I aptamers interact with the ligand via the 2′-hydroxyl groups of c-di-GMP, in contrast to class II aptamers [114,116,117]. Due to differences in the ligand recognition contacts, c-di-GMP analogs with bulky groups at the 2′ ribose, such as TBDMS, can still serve as class II riboswitch ligands. This discovery suggests that other chemical groups may be introduced at these positions to develop effective compounds that control gene expression in cells. Additionally, a group of 18 linear analogs was synthesized and tested for cyclic-di-GMP riboswitch aptamer binding and function. Linear analogs are more attractive for drug development; they are easier to synthesize and are less likely to bind to protein receptors, as they differ structurally from some circular derivatives of c-di-GMP. It was reported that all linear analogs interact effectively with the class II aptamer, and only six of these compounds were rejected by the more discriminatory class I aptamer at concentrations up to 100 μM. For both riboswitch classes, the natural breakdown product of c-di-GMP, pGpG, has the lowest *K*_D_ value (0.3 μΜ) among the linear analogs. When compared to analogs lacking the 5′ phosphate group, these compounds exhibit a relatively low difference in binding affinity for the class I riboswitch [119]. This is remarkable because both phosphate groups are expected to make contact with class I aptamer functional groups [116,117].

One of the twelve *C. difficile* class I riboswitch representatives was examined to promote in vitro transcription termination in the presence of each analog individually. The ligand and analogs interaction should trigger transcription termination. In the absence of the ligand, the full-length transcript was produced; however, when incubated with c-di-GMP or with the circular analogs 2 and 3, the reactions produced 50% terminated products. The rest of the tested analogs that produced lower binding affinities yielded substantially less termination. This demonstrates that the analog-binding kinetics or other structural differences between the analog–aptamer complexes affect the amount of transcription termination, rather than only *K*_D_ values. The “on”-switch c-di-GMP-II riboswitch of *C. difficile* was fully transcribed by analogs 2–5. The extent of full-length transcription is consistent with the circular analogs’ high binding affinities. Interestingly, the majority of linear analogs promoted the production of two-fold higher full-length transcripts compared to the riboswitch in the absence of any ligand [119].

Collectively, cyclic forms of analogs of the cyclic-di-GMP appear more effective than linear equivalents to target c-di-GMP riboswitches, despite the latter’s ease of synthesis. The exogenous injection of c-di-GMP and many of its analogs to bacteria promotes biological changes, including a reduction in the biofilm formation of *Streptococcus mutans*, *S. aureus* and *Pseudomonas aeruginosa*, according to various studies [120,121,122,123]. Although the mechanism of action of these compounds is unclear, it is proposed that when they enter cells, they operate intracellularly on c-di-GMP target molecules. Whether these compounds regulate bacterial behavior by interacting with the c-di-GMP riboswitch, and if analogs with alternative specific effects can be designed, remain important open questions for further investigation.

## 7. Lysine Riboswitches

Lysine riboswitches monitor intracellular lysine concentration and control the expression of the *lysC* gene; they are found in many Gram-positive and Gram-negative bacteria, including several clinically relevant pathogens. Th *lysC* gene encodes aspartokinase II, which catalyzes the first step in the conversion of L-aspartic acid to L-lysine [37]. 2,3-dihydropicolinate and L,L-diaminopimelate, two lysine intermediates in this metabolic pathway, are precursors for cell wall biosynthesis and spore production [124,125]. A second copy of the riboswitch is found in many bacteria and controls the production of a lysine-specific importer (coded by the *yvsH* gene in *B. subtilis*). Each lysine riboswitch aptamer domain’s secondary structure consists of five stem loops (P1–P5) radiating from a highly conserved single-stranded core [37,126,127]. The stems P2 and P3 base pair through their terminal loops, and P2 has both a loop E motif and a kink-turn (K-turn) motif [128]. In most bacteria, lysine binding stabilizes this structure, causes transcription termination and suppresses the expression of the downstream genes [37]. The riboswitch generates an alternative structure with an antiterminator hairpin at low lysine concentrations, allowing transcription. Due to the central role of lysine and its intermediates in key metabolic events, this riboswitch is considered an important drug target [129].

L-aminoethylcysteine (AEC) and DL-4-oxalysine were identified as inhibitors of gene expression by activating the lysine riboswitch, after screening lysine-related compounds [129]. These compounds bear a substitution on the carbon atom at position 4 by sulfur (AEC) and oxygen (oxalysine). Their affinities are similar to those of lysine (lysine 1 μM, AEC 30 μM, DL-4-oxalysine 13 μM) and both lead to bacterial growth inhibition [51]. AEC was first discovered in 1958 and was later reported to bind to the lysine aptamer in vitro, causing the downregulation of a reporter construct linked to the wild-type riboswitch, which could lead to the observed inhibition of the bacterial growth [37,130]. Moreover, AEC can be integrated into proteins, which is likely to be one mechanism leading to AEC-mediated toxicity [131,132]. In addition, AEC-resistant *B. subtilis* and *E. coli* strains were reported to bear a mutation in the *lysC* riboswitch [133,134]. All these observations led to the proposal that the suppression of the bacterial growth is at least partially attributed to AEC’s direct binding to the lysine riboswitch, leading to the suppression of aspartokinase II expression to a level that is harmful to cell growth [129]. However, later, the lysyl-tRNA-synthetase (LysRS) was characterized as the main target of AEC, while the lysine riboswitch is the key location for gaining antibiotic resistance. AEC resistance provided by riboswitch mutations results in a competition between lysine and AEC for LysRS binding, suggesting an indirect effect of the lysine riboswitch-mediated elevated levels of aspartokinase and the cellular concentration of lysine [135]. Future design efforts should identify compounds that target both LysRS and the lysine riboswitch [135]. The resolution of the crystal structure of lysine riboswitches in the presence of AEC and oxalysine revealed that the ligand binding pocket accommodates two openings at the C4 and N7 positions of bound lysine, and that both C4-substituted antibacterial chemicals fit into the relatively stiff binding pocket [136,137]. Their weak binding could be attributed to the substituent’s greater electronegativity at the C4 location [137]. These findings suggest that the C4 and N7 positions of bound lysine could be used for the design of novel and effective lysine analogs.

## 8. T-Box Riboswitches

T-box riboswitches represent a distinct class of riboswitches that were initially dis-covered as a novel class of tRNA-dependent transcription attenuators [25]. T-boxes are unique since they respond to the binding of tRNA molecules and today are known not only to regulate transcription, but also the translation of downstream genes. They sense amino acid availability by sensing the aminoacylation status of their tRNA ligands and are found principally in Gram-positive bacteria. T-boxes control the expression of aminoacyl- tRNA synthetase genes and genes involved in amino acid biosynthesis and uptake [7,25,138,139]. The first domain of T-boxes is stem I, which includes distinct structural elements such as the kink-turn (K-turn), the AG bulge, the apical loop and the essential specifier loop, which contains a codon-like triplet that base-pairs with the tRNA’s anticodon. This sequence-specific interaction between the specifier sequence and the tRNA anticodon ensures the T-box:tRNA complex establishment and is secured through an essential local geometry which is favored by flanking stacked bases (conserved purines located near the specifier loop codon or tRNA anticodon triplet) [140]. Subsequently, specific identity elements of tRNA’s overall structure favor essential dynamic associations in order to secure the correct accommodation of the ligand, leading to the interaction between the apical loop and the tRNA elbow (G19 of the D-loop and C56 of the T-loop) [141,142]. Following stem I, a linker sequence can be discerned which can host a highly structured stem II domain, which serves to stabilize stem I–tRNA interactions. These structural features are followed by the discriminator domain (also termed the antiterminator or antisequestrator domain), which houses the conserved T-box bulge sequence which base pairs with tRNA’s conserved 3′ CCA end [143]. This interaction occurs between the highly conserved 7-nucleotide sequence of the bulge and the 3′ CCA acceptor stem of the tRNA, which stabilizes the antiterminator/antisequestrator conformation and senses the aminoacylation status of the bound tRNA. Based on recent cocrystal and cryo-EM structures of Geobacillus kaustophilus, Bacillus subtilis and Mycobacterium tuberculosis T-box-tRNA complexes, the T-box envelopes the tRNA, capturing its 3′ end, and positions it to face a steric barrier in the structure created by a steric filter (G167-U185). The steric sieve rejects any aminoacyl group that is present on the tRNA 3′ end, and conditionally licenses a functionally crucial stacking interaction between the T-box and the tRNA (C186::tA76) only when the tRNA is uncharged [144]. Therefore, the conformational switch between the termination and antitermination of transcription or between the initiation and blockage of translation is determined by whether the bound tRNA is charged with its corresponding amino acid or not, which is directly sensed by the antiterminator/antisequestrator domain in conjunction with immediately adjacent structural features such as the stem III. Thus, T-boxes can sense the aminoacylation status of tRNAs in both transcription and translation, making them important sensors of amino acid availability, and thus, major regulators of bacterial survival and adaptation in different metabolic environments [7,144,145,146,147]. Due to their complex tertiary structures, T-boxes are considered attractive drug targets against Gram-positive bacteria, including a number of prominent human pathogens such as Bacillus, Staphylococcus, Streptococcus, Clostridium, Mycobacterium, Listeria and others [148].

Given the central role of T-boxes in the cellular metabolism, several approaches were used to develop antibacterial compounds that could selectively bind to a distinct domain of the T-box (Figure 3). In a first attempt, the functionally relevant T-box antiterminator element model, AM1A, from *B. subtilis tyrS* T-box was used. AM1A is fully functional in vivo when used in conjunction with the full-length leader sequence [149]. This model was used for the investigation of the binding of eight different aminoglycosides (neomycin B, kanamycin A, kanamycin B, amikacin, tobramycin, paromomycin, gentamicin C and streptomycin) through fluorescence resonance energy transfer (FRET). The antibiotics binding affinity with the RNA ranges from low- to mid-μM values, showing that small molecules can selectively bind to T-box antiterminator RNA [150]. The number of amines of the aminoglycosides studied correlates with their binding affinities. More specifically, neomycin exhibited the lowest dissociation constant among the eight aminoglycosides evaluated, which is in line with previous studies showing that neomycin binds to RNA with the highest affinity among monomeric aminoglycosides [151,152,153,154] (Table 1). Paromomycin, gentamicin C and kanamycin B appeared to have significant binding, in contrast to kanamycin A and streptomycin. Previous studies have shown that aminoglycosides bind well to bulged bases and RNA pockets, notably in divalent metal ion binding sites [155]. Aminoglycosides most likely interact with the antiterminator RNA model via electrostatic interactions [152,154,156,157]. The antiterminator RNA forms a unique RNA pocket that is amenable to selective small molecule binding, explaining the 93-fold selectivity difference between the tightest-binding aminoglycoside (neomycin B) and the weakest (streptomycin), as well as the four-fold selectivity difference between tobramycin and kanamycin B (both containing five amines) [150]. Later studies showed that the neomycin B binding site in the T-box antiterminator is specifically located at the 5’ end of the bulge, indicating that the bulge nucleotides create a ligand-binding pocket that might possibly also be a divalent metal ion-binding site. Notably, the electrostatic interaction promotes and does not prevent the tRNA binding in the antiterminator region [69].

Another study that also used the AM1A model identified 4,5-disubstituted oxazolidinones that bind to the T-box antiterminator RNA element and impact antitermination directly. Although these were amine-substituted analogs of oxazolidinones, the basic amine was included to significantly improve their binding affinity. One of them (compound 2c) served as an inhibitor of the tRNA-mediated antitermination in vitro, a result attributed to its high affinity and structural specificity. On the other hand, another similar compound (2d) promoted in vitro antitermination independently of the presence or the absence of tRNA, suggesting that this compound stabilizes the antiterminator, thus, reducing the requirement for tRNA binding. A FRET analysis, as well as enzymatic probing experiments, verified that compounds 2c and 2d exhibit different binding modes, explaining the different effects on antitermination [158,159]. Furthermore, a class of conformationally restricted oxazolidinone-triazole compounds that target the T-box riboswitch’s antiterminator RNA element was designed and synthesized [160]. T-box riboswitch function, but not transcription in general, was hindered by the conformationally restricted compounds. Some of these compounds successfully blocked the riboswitch readthrough, although there was little to no inhibition of the tRNA–antiterminator complex creation. Data from complex disruption, computational docking and RNA binding specificity studies suggest that the compounds may negatively modify the conformation of the tRNA–antiterminator complex, instead of disrupting its formation via competitive inhibition, thus, functioning as negative allosteric modulators of T-box riboswitch activity. This allosteric activity can be achieved by binding to an allosteric site, a site that is different than where tRNA directly contacts the antiterminator, to possibly impair the T-box antiterminator element function [160].

Recently, PKZ18, a small-molecule with an inhibitory effect against several T-boxes and with significant antibacterial activity against both Gram-positive and Gram-negative bacteria was identified [70]. PKZ18 emerged after an extensive in silico docking analysis of more than 300,000 compounds, which was narrowed down to 700 compounds with potential binding affinity to the specifier loop of *B. subtilis tyrS* T-box. Among those, PKZ18 was selected as the best hit due to the specificity of its antibacterial activity against Gram-positive bacteria, its inability to produce resistance in diverse bacterial strains and its ability to bind the specifier loop in vitro. Subsequent in vitro experiments suggested that PKZ18 binds to the specifier loop of both *tyrS* and *glyQS* T-boxes. In addition, in vivo experimentation showed that PKZ18 inhibited both the *glyQS* transcription and TyRS protein production, suggesting that PKZ18 acts against multiple T-boxes. The resistance of this compound arose at a low mutational frequency of 1.21 × 10^−10^, due to the broad binding of different T-boxes. Moreover, after a 24-h exposure to PKZ18, no substantial cytotoxicity was found to any of the eukaryotic cell lines tested at the bacterial minimum inhibitory concentrations (MIC) [70]. In a next step, improved analogs of PKZ18 were characterized that exhibit better MICs and bactericidal activity against MRSA, and when compared to PKZ18, showed decreased cytotoxicity against eukaryotic cells [71]. Interestingly, the combination of PKZ18 analogs with aminoglycosides improved the effectiveness of both and broadened their therapeutic window. More specifically, analog PKZ18-22 exhibited a significant effect on the expression of eight out of 12 T-box regulated genes on MRSA and did not affect the 5′ UTR of other genes, as revealed by RNA sequencing. It was found that PKZ18-22 has a wider range of biofilm-inhibiting properties and is 10 times more effective than frequently used antibiotics such as vancomycin [72]. Additionally, the very low levels of resistance to PKZ18 analogs supports the specific binding of them on different T-boxes and represents attractive small-molecule antibacterial drugs for future research and clinical use due to their diversified T-box targets, their minimal occurrences of resistance and their synergy with other antibiotics [71].

Recently, it was demonstrated that mainstream protein synthesis inhibitor antibiotics can directly regulate T-box controlled transcription antitermination [28]. More specifically, neomycin B, pactamycin, paramomycin and tigecycline were found to enhance the *S. aureus glyS* T-box mediated transcription, whereas chloramphenicol and linezolid to reduced it both in vivo and in vitro. Interestingly, neomycin B and tigecycline also appear to bind to the D-loop region of tRNA^Gly^, near the anticodon loop’s wobble position and at the anticodon stem and D-arm stacking interface. The binding sites of linezolid, neomycin B and tigecycline on the *S. aureus glyS* T-box were analyzed by chemical probing and showed that the binding sites of neomycin B and tigecycline appeared almost identical on the specifier loop, the AG bulge and the apical loop of stem I, as well as on the staphylococcal-specific stem Sa contributing to the T-box:tRNA complex stability. Although linezolid exhibits binding positions on the specifier loop, the AG bulge and the apical loop of stem I, no binding sites were observed on stem Sa [28]. Notably, when stem Sa was deleted from the *S. aureus glyS* T-box, tigecycline inhibited the *glyS* T-box mediated transcription in vivo, whereas linezolid retained its inhibitory effect. Both inhibitors appeared to bind to the linker and stem III when stem Sa was absent, but only linezolid appeared to compete with the tRNA for binding to the same sites of stem I. Furthermore, tigecycline could bind the linker sequence and the cap of the antiterminator stem, whereas linezolid at the base of stem III. In the presence of tigecycline, stem Sa insertion in the terminator/antiterminator domain of *Geobacillus kaustophilus glyQ* T-box, which naturally lacks this domain, led to increased transcription as well. A probing analysis showed that tigecycline induced protection on multiple previously known and new sites, a discovery that explains why the mutant of *G. kaustophilus glyQ* T-box appended with stem Sa increased in vivo the transcription readthrough in the presence of tigecycline. In conclusion, *Staphylococcal*-specific stem Sa can serve as a potential lineage-specific drug target since it facilitates the binding and function of protein synthesis inhibitors [30]. These recent observations, taken together, strongly suggest that targeting T-box riboswitches can be both species-specific and effective, thus, providing a proof-of-concept for the development of new inhibitors that interfere with specific RNA structural features.

## 9. Discussion

Since the discovery of riboswitches over 30 years ago, more than 40 riboswitch classes have emerged and been characterized as evolutionarily conserved components of bacterial RNA-mediated regulation [38,74,161,162]. Riboswitches offer new possibilities for controlling gene expression in cells. Synthetic riboswitches have already been designed to inhibit or activate gene expression in a ligand-dependent manner, just like natural riboswitches. Despite the already existing systems of gene regulation, the riboswitch-based system characteristics add new value to this field. Riboswitches are modulated by relatively small and simple synthetic ligands, thus, decreasing the costs of widespread usage, even at an industrial scale. Additionally, drug administration in the growth medium is sufficient to induce the desirable regulatory effects, as the ligands are often small enough to pass through the cell wall and membrane. Moreover, riboswitches offer great variety, use a broad range of regulatory mechanisms and the ligand binding can result in the up- or down- regulation of gene expression, making them versatile tools for controlling a wide range of genes [163].

Synthetic riboswitches are designed by utilizing either a natural riboswitch to regulate the expression of the desired gene, or the combination of different riboswitches, or by identifying new aptamers in vitro [164] (Table 2). However, the use of aptamers derived from natural existing riboswitches to recognize a synthetic ligand is restricted by the fact that the endogenous natural ligand can result in background activity. In order to overcome this limitation, natural aptamers should be modified to specifically recognize the synthetic ligand, while the expression platform remains unaltered. Those aptamers in combination with different expression platforms have also presented high variability. Except for the utilization of naturally occurring riboswitches, the in vitro systematic evolution of ligands by exponential enrichment (SELEX) is also used for the identification of aptamers that bind non-natural ligands [165]. However, the conformational changes that are required to make the aptamer active after ligand binding and the artificial conditions under which the aptamers are selected may hamper their functionality in vivo. Furthermore, the expression platform is more complex to engineer. For instance, the translation is differently regulated in bacteria than eukaryotes. So, in eukaryotes the riboswitch does not need to regulate the RBS for the ribosome, but the scanning process of the ribosomal subunit or the IRES folding. By overcoming these barriers, engineered riboswitches managed to halt translation both in a cap- and IRES-dependent manner, as well as by controlling ribosomal shunting and splicing. Further research has also shown that gene expression can be regulated from synthetic riboswitches through RNA self-cleavage, by interfering with the RNAi pathway or by controlling microRNA (miRNA) production [164,165,166,167].

The artificial theophylline-responding riboswitch was developed in 2012 as a model for tuberculosis research, in order to modulate gene expression in a variety of Gram-positive and Gram-negative bacteria [168]. Specifically, this system is based on a synthetic aptamer domain that detects theophylline and a mycobacterial promoter with a length of around 300 nt, and it was used for a conditional gene knockdown, for the regulation of gene expression in a macrophage infection model, and to reversibly induce or inhibit heterologous protein expression. In the absence of theophylline, the riboswitch adopts an “off” conformation, which blocks the RBS and start codon for translation initiation. The riboswitch construct in *Mycobacterium smegmatis* and *Mycobacterium tuberculosis* induced the expression of GFP and β-galactosidase in a dose-dependent manner. The advantage of this system is that it is encoded by a DNA sequence upstream of the target gene and does not necessitate any accessory proteins. In addition, theophylline is an FDA-approved medicine and it is well tolerated in mice and guinea pigs, which would enable the prospective application of the theophylline-responsive riboswitch system to animal models [168].

Moreover, riboswitches provide an attractive new strategy to the biosensor field [169]. When the aptamer of the synthetic riboswitches is coupled to GFP or is fluorescent, it can sense the availability of cellular metabolites, such as glycine and cyclic di-AMP [170,171]. In 2009, an anti-asthmatic drug theophylline biosensor was developed [172]. Since theophylline overdose can be harmful to health by inducing seizures and cardiac rhythm problems, monitoring theophylline content in the blood is crucial [173]. The constructed biosensor consisted of a theophylline binding aptamer domain (acting as a sensing element) connected to the 5′ end of a GFP coding sequence. The signal transducer was able to transform the dose-dependent stimulation of GFP expression mediated by theophylline’s interaction with the riboswitch into quantifiable digital data. The identification of several ligand analogs provides the potential for widespread application in the medical industry. The ongoing structural characterization of riboswitches gave rise to the development of new high throughput approaches to discover and identify new ligands that specifically target the riboswitch aptamers. A few of these strategies have already identified specific compounds that mimic the natural ligand and exhibit antimicrobial activity. However, additional approaches have been adopted and characterized compounds that are not tested for their antimicrobial activity, yet. For instance, a small molecule microarray (SMM) screen enabled the identification of a compound that binds to *B. subtilis preQ1* riboswitch (*K*_D_ ~500 nM). The compound regulates riboswitch function, despite its interaction with different nucleotides compared to the natural ligand (PreQ1) and induction of a different riboswitch conformation upon binding [174]. A fluorescence polarization (FP) assay identified a compound [18], which was developed by inserting a functionalized linker at different positions of the SAM ribose and exhibited different binding profiles towards *B. subtilis* and *S. aureus* SAM-I riboswitches [175]. Such fluorescence strategies have been widely used to identify the novel ligands of riboswitches, as they can be automated and rapidly reveal ligand–riboswitch interactions [176,177,178]. In addition, computational approaches that rely on the riboswitch sequence and structure can help to determine the suitability of a riboswitch aptamer to interact with a noncognate ligand, and thus, apply the appropriate modifications to induce the riboswitch’s conformational change [179,180]. The examples described herein show that riboswitches have drawn scientific and industrial attention as alternative antibacterial targets, and that a variety of effective compounds are already available. Only one of these—targeting the guanine riboswitch of *S. aureus*—has so far been used in animal research [67], despite some reported activity in in vivo bacterial growth experiments. However, there are still some challenges that need to be addressed for the development of the ideal compounds. A promising compound should approach or surpass the binding affinity of the natural ligand and should not interact with other, unintended metabolic pathways. In the majority of known cases, the antimicrobial effect of the so far studied compounds results from the downregulation of a single gene or an operon by a single targeted riboswitch, which allows for a relatively fast development of resistance. Additionally, since riboswitches are absent in humans and in higher eukaryotes, a possibility of unintended toxicity of the metabolite analogs always exists. Therefore, the compounds should be designed to specifically target the riboswitch and avoid having any off-target effects on human proteins. Taking into consideration that riboswitches generally recognize their ligand differently than proteins, this is reasonably achievable [26,51].

The discovery, development and implementation of additional applications of riboswitches are ongoing and gaining momentum [181]. Bacterial gene-regulatory RNAs in general and riboswitches in particular appear as ideal molecular tools and antibacterial drug targets. Advances in understanding their molecular mechanisms are of great interest and have only just begun to unveil their full potential for the development of a new world of specific and effective antibiotics to combat the ongoing pandemic of antibiotic resistance.

## Figures and Tables

**Figure 1 antibiotics-11-01243-f001:**
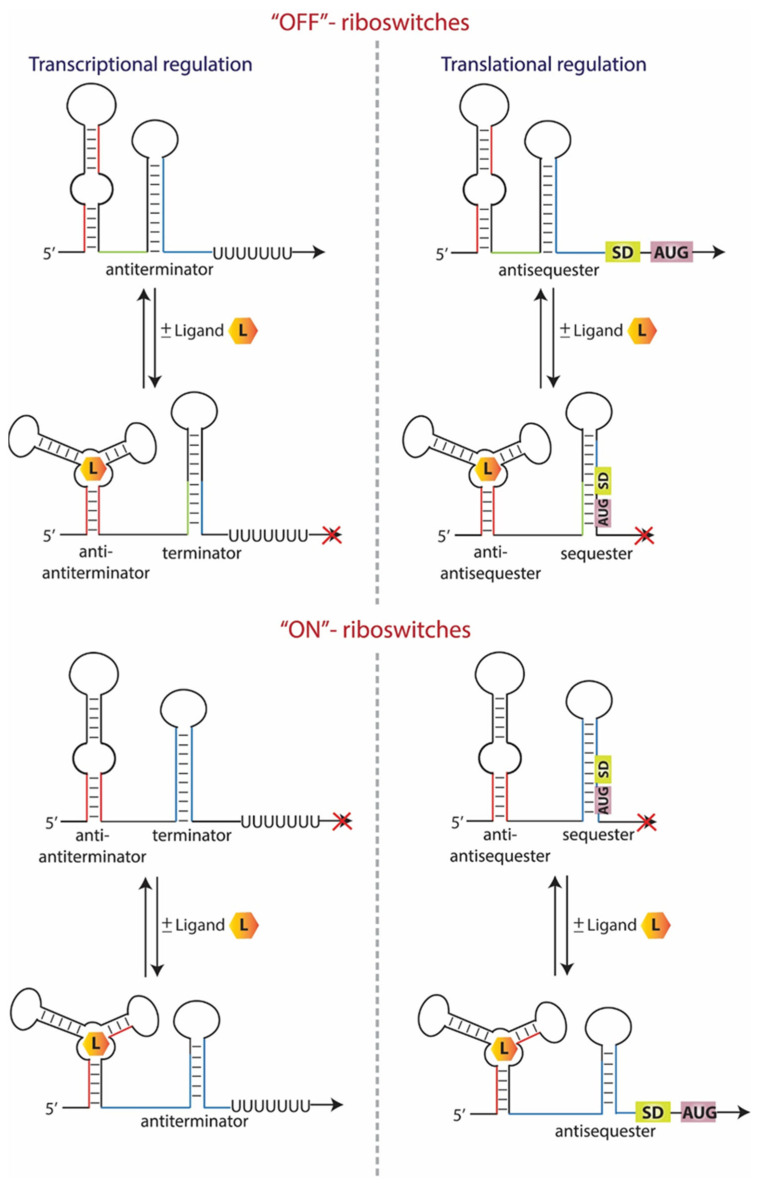
Gene regulation by riboswitches. The binding of the cognate ligands results in termination of transcription (**left**) or translation (**right**) of the downstream genes in the “OFF” riboswitches. In contrast, “ON” riboswitches bind their ligand and modulate their conformation to enable the expression of their downstream genes.

**Figure 2 antibiotics-11-01243-f002:**
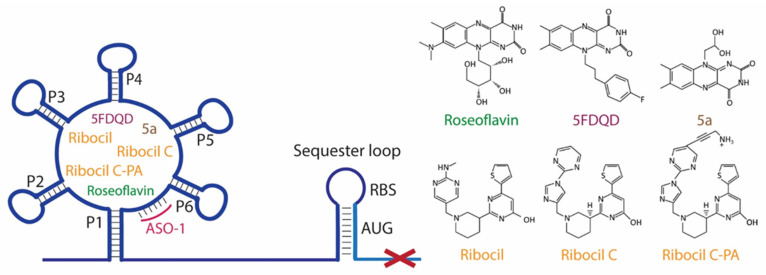
Schematic representation of FMN riboswitch. Different antimicrobial compounds bind to the FMN aptamer and abolish the expression of the downstream genes (**left**). The chemical structure of the antimicrobial compounds is also indicated (**right**).

**Figure 3 antibiotics-11-01243-f003:**
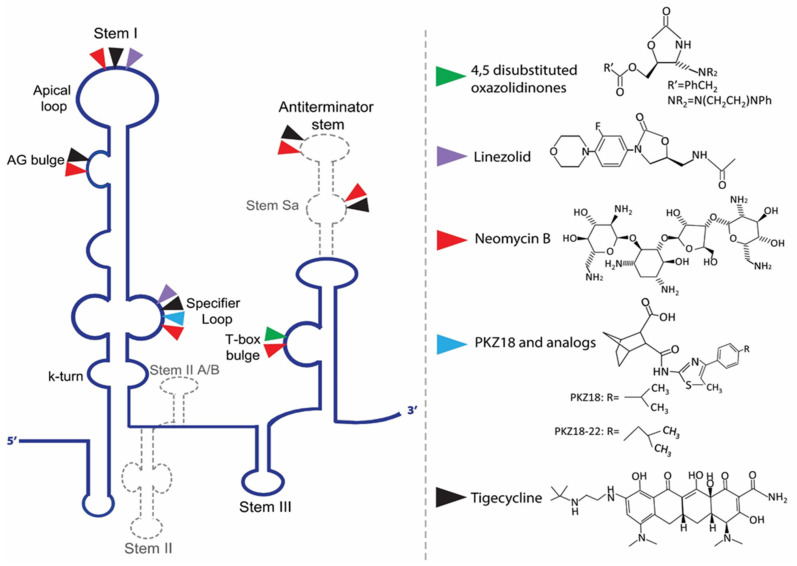
Illustration of the proposed model of antimicrobial compounds binding sites on the T-box riboswitch (**left**). The arrows indicate the binding positions of the antimicrobial compounds that target the T-box riboswitch. The stems that are not always present on the T-boxes are illustrated with dashed lines. Each color of the arrows represents a different compound. The chemical structure of the antimicrobial compounds is also indicated (**right**).

**Table 2 antibiotics-11-01243-t002:** Characteristics of natural and synthetic riboswitches.

	Natural Riboswitches	Synthetic Riboswitches
Signals	Metabolites, ions, synthetic ligands	
Regulated mechanism	Transcription, translation	Transcription, translation, RNAi pathway, RNA self-cleavage
Structural modules	Aptamer domain and expression platform	Aptamer–aptamer domain and expression platform, two riboswitches, aptamer domain and ribozyme, aptamer domain and expression platform from different riboswitches
Applications	Gene expression regulation	Small molecule reporters, conditional gene regulation, metabolic flux engineering, fluorescent biosensors

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
