# Peer review of "A Riboswitch-Driven Era of New Antibacterials"

_antibiotics, 2022, doi:10.3390/antibiotics11091243_

Round 1
Reviewer 1 Report
- The authors stated Riboswitchs, However, more details in shorter clear paragraphs are needed so that the reader can understand the impact of this information.
- Always a graphical abstract at the end of the Discussion may further makes the paper more reader- friendly.
- The clarity of the presentation needs to be improved.
- Rephrase and explain the Discussion in shorter clear paragraphs will be more reader-friendly
Author Response
We thank the reviewer for the comments. The revised manuscript includes further information, especially in the discussion section. In addition, the text was proofread by native English speaker to address issues in expressions that required editing throughout the text. We hope that the revised manuscript appears now improved.
Reviewer 2 Report
Summary
The focus of the review is to summarize the structural noncoding RNAs called riboswitches, found in 5’ UTR of important genes. Riboswitches regulate gene expression upon binding of various ligands such as ions, nucleotides and tRNAs and affect downstream transcription or translation due to change in their mRNA conformations. Due to their regulatory role, riboswitches have been exploited as novel RNA-based targets for the development of new generation antibacterial drugs that can overcome impending drug-resistance problems. Herein, authors provide a comprehensive summary of and on the available knowledge of known compounds that target several main riboswitches such as TPP, FMN, GlmS, Guanine, Cyclic-di GMP, lysine, and T-box riboswitches and discuss their role as a novel class of inhibitors.
This is a very well written review on riboswitches however it is missing a section on synthetic riboswitches and what is potentially known in the field in terms of design strategy, applications in terms of design and target flexibility in comparison to naturally occurring riboswitches.
Authors should include a section on synthetic riboswitches and elaborate on the following:
- Definition and current understanding of the domain
- potential design strategies (for eg. In the 3’UTR before polyA stretch etc.)
- a comparative table between naturally occurring vs synthetic riboswitches with a focus on the applications.
Author Response
We thank the reviewer for the comments and suggestions. According to the reviewer's suggestions we expended the discussion section for the synthetic riboswitches and we included a small table with th major characteristics between natural and synthetic riboswitches, especially regarding potential applications.
We hope that the revised manuscript now appears improved. We would like once more to thank the reviewer for the insightful comments and the helpful suggestions.
Reviewer 3 Report
The authors organized a nice review on a select group of riboswitches that are promising for the development of a new generation of antibacterial drugs. The topic is relevant, important , and of great interest to the readers in this field. I believe it will motivate more researchers to develop novel inhibitors that target important bacterial regulatory RNAs. The references in this review are appropriate, thorough and up-to-date.
It is a good contribution to the readers in this field. I recommend accept in present form for publication.
Author Response
We thank the reviewer for the kind comments on our manuscript and we are happy to see that our manuscript covered the current field in the best way possible.